# Self-Interested Coalitional Crowdsensing for Multi-Agent Interactive Environment Monitoring

**DOI:** 10.3390/s24020509

**Published:** 2024-01-14

**Authors:** Xiuwen Liu, Xinghua Lei, Xin Li, Sirui Chen

**Affiliations:** College of Computer Science and Technology, China University of Petroleum (East China), Qingdao 266580, China; xwliu2015@whu.edu.cn (X.L.); z23070092@s.upc.edu.cn (S.C.)

**Keywords:** multi-agent reinforcement learning, self-interested coalition crowdsensing, environment monitoring, hidden confounder, worker selection

## Abstract

As a promising paradigm, mobile crowdsensing (MCS) takes advantage of sensing abilities and cooperates with multi-agent reinforcement learning technologies to provide services for users in large sensing areas, such as smart transportation, environment monitoring, etc. In most cases, strategy training for multi-agent reinforcement learning requires substantial interaction with the sensing environment, which results in unaffordable costs. Thus, environment reconstruction via extraction of the causal effect model from past data is an effective way to smoothly accomplish environment monitoring. However, the sensing environment is often so complex that the observable and unobservable data collected are sparse and heterogeneous, affecting the accuracy of the reconstruction. In this paper, we focus on developing a robust multi-agent environment monitoring framework, called self-interested coalitional crowdsensing for multi-agent interactive environment monitoring (SCC-MIE), including environment reconstruction and worker selection. In SCC-MIE, we start from a multi-agent generative adversarial imitation learning framework to introduce a new self-interested coalitional learning strategy, which forges cooperation between a reconstructor and a discriminator to learn the sensing environment together with the hidden confounder while providing interpretability on the results of environment monitoring. Based on this, we utilize the secretary problem to select suitable workers to collect data for accurate environment monitoring in a real-time manner. It is shown that SCC-MIE realizes a significant performance improvement in environment monitoring compared to the existing models.

## 1. Introduction

With the explosion of wireless communication and portable mobile devices, mobile crowdsensing (MCS) [1,2] has become a popular paradigm that appeals to workers to implement various sensing tasks and provide recommendation services [3,4]. However, different sensing environments and requirements make traditional MCSs very time consuming in accomplishing sensing tasks. Consequently, a highly effective tool, known as multi-agent reinforcement learning [5,6,7,8,9,10,11], has been developed to handle sequence recommendations in MCS. This tool demonstrates remarkable potential for addressing decision difficulties in unfamiliar sensing contexts. However, training in multi-agent reinforcement learning requires significant interactions and costs, which can lead to a decrease in data efficiency. Therefore, it is impractical to interact directly with the sensing environment. According to recent studies [12], environmental reconstruction is a viable technique that utilizes imitative learning to acquire environmental strategies from historical data. This approach not only increases the efficiency of interacting with virtual environments but also decreases interaction costs.

However, real-world scenarios exhibit a high level of complexity, so the data collected by workers from different sensory environments is significantly multi-source and heterogeneous [8,9,10,11]. Figure 1 shows the variability of data collected in different sensing environments at different times. For example, data collected by workers from the target area tends to hold more significance compared to data collected from peripheral areas. Additionally, newly acquired data prove more valuable for environmental reconstruction than older data collected within the same sensing environment. Therefore, how to judge the importance of data and utilize it effectively is the first challenge of this paper.

Note that it is difficult for workers to collect fully observable data in real time [13,14,15,16]. Most collected data have hidden confounding factors [17,18]. In addition, the next state relies on the former state and the action performed during training for multi-agent reinforcement learning. In real-world situations, however, the next state is more likely to be additionally influenced by hidden confounders. Similarly, hidden confounders affect the agents’ actions and rewards as they interact with the sensing environment, which can reduce the accuracy of the environment monitoring. Hence, the second challenge is how to perform environmental monitoring in the presence of confounders.

Furthermore, rather than reactively waiting for given data, we are more interested in selecting the right workers to proactively sense important data for precise environmental monitoring. However, it is often complicated to forecast real-time data on workers in various sensing environments. Thus, the third challenge is how to select a group of workers to sense essential data in a real-time manner for accurate environmental monitoring.

To address the aforementioned difficulties, we have developed a unique framework called Self-interested Coalitional Crowdsensing of the Multi-agent Interactive Environment Monitoring (SCC-MIE). The objective of this framework is to acquire knowledge about the sensing environment and the underlying confounding factors utilizing partially seen data. It focuses on environment reconstruction and worker selection based on data confidence. In particular, we consider a reconstructor that comprises three representative agents: platform agent $\pi_{a}$, worker agent $\pi_{b}$, and confounding agent $\pi_{h}$, which leverages the available partial observations and historical data to reconstruct the missing or unobserved information. It aims to accurately model and understand the underlying structure of the sensing environment. Another discriminator is learned as a critic, outputting confidence values and judging whether the data samples are reliable or not. The reconstructor also uses the critic’s judgment to improve its performance and challenges the critic by providing as little information as possible. Finally, we invert the selection of workers based on the confidence of the data in the environmental reconstruction, which in turn promotes continuous improvement in reconstruction accuracy. By forging cooperation between these two tasks, SCC-MIE could maximize the accuracy and reliability of the environment monitoring:We propose an efficient framework, called SCC-MIE, which consists of multi-agent imitation learning and a secretary-based online worker selection strategy. Based on data spatiotemporal heterogeneity and confounding effects, the former estimates the importance of the data and reconstructs the sensing environment. The latter aims to select workers to proactively sense critical data in an online manner to motivate environmental reconstruction.Considering confounding effects in real sensing environments, we design an imitation learning framework that includes confounder-embedded policy and a discriminator to learn the policy based on their interactions effectively.Extensive Evaluation: we conducted an extensive evaluation of the dataset using four different methods, which verified the validity of SCC-MIE.

## 2. Related Work

Mobile crowdsensing is a popular pattern that exploits groups of workers to perform a variety of sensing tasks in large-scale areas. To efficiently collect data, researchers have proposed numerous MCS schemes. For instance, in reference [19], the method employed dynamically priced tasks and social network effects to incentivize worker participation in MCS campaigns. Reference [20] used a set of UAVs to collect data from multiple mobile users to maximize geographic coverage and reduce the age of information (AoI) for all workers. Furthermore, crowdsensing platforms utilized polynomial-time greedy algorithms in task assignments to ensure fairness and energy efficiency among participating workers [4]. However, most existing works have overlooked the fact that the sensing data submitted by workers is often sparse and incomplete. As research progresses, compressed perception and matrix completion have emerged as practical options for data supplementation [10,11]. Compressed perception techniques recover the original data by observing low dimensional features, whereas matrix completion methods fill in missing values based on data structure and sample correlation. Applying these methods improves the handling of sparse and incomplete perceptual data.

Nevertheless, the present research has not thoroughly investigated the spatiotemporal correlation of perceptual data. In MCS environments, sensing data exhibit spatiotemporal correlations, wherein data from neighboring points in time and space may interact with each other. Therefore, considering this spatiotemporal correlation becomes crucial when performing state estimation to minimize resulting errors.

To address these issues, several recent studies have used reinforcement learning to establish a direct connection between spatiotemporal regions and their corresponding accuracy, to estimate their significance. Unfortunately, it is costly to run RL algorithms, which require much interaction with the environment to improve accuracy. As a result, environment reconstruction in RL has attracted widespread attention recently. Various studies have shown that imitation learning [21,22] enables environmental reconstruction by learning environmental strategies from past data. For instance, generative adversarial imitation learning (GAIL) [6] and its extension [23], multi-agent imitation learning (MAIL), simultaneously learn both strategies by defeating the discriminator that discovers the difference in the generated data from the real data. Furthermore, these workers are not directly applicable in practical situations. This is because the real world cannot provide a perfect sensing environment [14]. There is a high likelihood of confounders that affect the accuracy of the environment reconstruction [17].

Therefore, we present an efficient method that enables real-time environment reconstruction, keeping the importance estimates up to date, and selecting workers who can actively sense important data for environment reconstruction. The framework will combine reinforcement learning and imitation learning approaches to achieve accurate environment reconstruction in complex and incompletely observable real-world environments. In this way, we can overcome the challenges posed by the high cost and real-world complexity of traditional RL algorithms and provide an effective solution to environmental reconstruction in mobile crowdsensing.

## 3. Problem Definition and Framework Overview

### 3.1. Problem Definition

In this section, we define the MCS scenario and the sensing data. Then, our research question, worker selection, is determined, which leads to the focus of the article, namely environmental reconstruction. In a typical MCS campaign, each sensing task should be executed in m target sensing regions with a duration of T. In this paper, we divide the duration T into t periods short enough to keep the sensed data stable and real time. Moreover, the main notations of this paper are listed in Table 1.

Next, we define the trajectory and sensing data. The research question of this paper is defined according to them.

**Definition** **1.**
*(Trajectory) A trajectory is a chronological sequence of spatiotemporal points. In addition, n workers collect trajectories in time T, denoted as $\tau =\{\tau_{1},\tau_{2},\ldots ,\tau_{n}\}$, where each point $\tau_{i}=\left(x,y,t\right)$ is made up of positions (x,y) (i.e., longitude and latitude) at the t-time slot.*


**Definition** **2.**
*(Sensing Data) For each sensing task, we consider a set of workers, denoted as $w_{i}=\{w_{1},w_{2},\ldots ,w_{n}\}$, each worker i ∈N with a cost $c_{i}$ independently determines its sensory data $X_{i}^{t}$ at t-time slot.*


We assume a set of tasks with m sensory regions, n time periods, a budget of B, and a duration of T. In this paper, our primary objective revolves around the precise reconstruction of the environment while minimizing the discrepancy between the generated data and real-world data. To achieve this goal, we focus on selecting workers who will contribute their individual sensing data to the MCS campaign. The decision-making process for the platform involves assessing the estimated importance of each worker’s data and considering the remaining budget, denoted as B, to determine whether to include a worker’s data for the task of environment reconstruction.
(1)$$minimize\sum_{t=1}^{T}D\left(X_{t},\hat{X_{t}}\right)={\left(X-\hat{X_{t}}\right)}^{2}$$
(2)$$subject\;to\;\hat{X_{t}}=G\left(X_{t}\right),\;w\subseteq W,\;\sum_{w_{i}\in w}c_{i}\leq B$$

In our analysis, we assume that the addition of new data leads to a reduction in error and transforms the objective function into a maximization problem. However, the challenge lies in the worker selection problem, which essentially involves choosing a subset of k elements to maximize an ensemble function [24]. Taking into account the intricate nature of environment reconstruction and the dynamically evolving real-time scenarios, we recognize that the main problem addressed in this paper falls under the realm of NP-hard problems.

### 3.2. Framework Overview

In this paper, we propose the SCC-MIE, which consists of two key components: worker selection and environmental reconstruction. Figure 2 illustrates the overall architecture of SCC-MIE, showing the interaction between these two key elements.

Worker Selection: this component determines whether or not to select the worker based on the quality of the data. Firstly, sampling from historical trajectories. Then, importance estimation is derived from the spatiotemporal data of the sample (workers). Ultimately, a secretary strategy is adopted to select suitable workers, aiming to sense important data for environment reconstruction proactively.

Environment Reconstruction: we develop an online framework for reconstructing the environment using spatiotemporal input from workers. This framework consists of a reconstructor and a discriminator. Furthermore, SCC-MIE forges cooperation through information sharing among agents to boost their performance. First, the reconstructor extracts the spatiotemporal embedding of the data and then reconstructs the environment by considering the ignored confounders. The discriminator receives additional information from the reconstructor and computes confidence measures for the estimation results by observing the reconstructed data.

## 4. Online Multi-Agent Environment Reconstruction

Due to the hidden confounding factors in realistic sensing environments, we introduce a novel online multi-intelligent environment reconstruction method, namely SCC-MIE (self-interested coalitional crowdsensing for multi-agent interactive environment monitoring). Our approach aims to tackle two interconnected tasks simultaneously. The first task involves utilizing a reconstructor G with generative capabilities to perform environmental reconstruction by generating an approximation $\hat{X}$ based on partial observations X. The second task leverages a discriminator D to calculate an interpretable confidence level P, incorporating information from X and $\hat{X}$. This parallel undertaking enables us to solve the complexities of environment monitoring effectively in the presence of hidden confounders.

The two tasks can be formalized as follows:(3)$$A:G\left(X\right)=\left(\hat{X}\right),{ℒ}_{A}=loss_{A}\left(X,\hat{X}\right)$$
(4)$$B:D\left(X,\hat{X}\right)={\left(X-\hat{X}\right)}^{2}=P$$

Considering the cooperative and interactive nature of the two tasks defined in Equations (3) and (4), an intuitive solution is adversarial learning, such as a generative adversarial network (GAN), which not only enables mutual learning between the generator and the discriminator but also improves the performance of both tasks. As a result, we want to minimize the following loss function:(5)$$\underset{G}{\mathrm{argmin}}\arg\max_{D\in 0,1}E_{X\sim p_{E}}[\log D(x)]+E_{z\sim p_{Z}}[\log(1-D(G(z)))]$$
where $p_{z}$ is a distribution, the generator is responsible for generating samples (in our case, $\hat{X}$) that approximate the desired data distribution $p_{E}$. However, it is essential to note that the training of the primary task heavily relies on the output of the discriminator D, which can result in the potential loss of essential information. Furthermore, solving the minimax optimization problem involved in GAN-style models is inherently more challenging compared to directly minimizing the loss function.

In addition, we explore an alternative approach called generative adversarial imitation learning (GAIL), which has gained popularity as a method for imitation learning. Unlike traditional imitation learning, which directly learns policies from expert demonstrations and has shown practical benefits, GAIL takes a different approach. It leverages the concept of generative adversarial networks to learn policies by competing against a discriminator network trained on both expert demonstrations and generated trajectories. This approach offers a distinct perspective on imitation learning and has shown promise in various applications.

The problem at hand can be formulated as follows: we aim to train a policy π that minimizes the loss function l(s,π(s)) under the discounted state distribution of the expert policy $P_{\pi_{e}}\left(s\right)=\left(1-\gamma \right){\sum_{t=0}^{T}\gamma }^{t}p\left(s_{t}\right)$. The objective of imitation learning is denoted as $\pi =\arg\min E_{s∼P_{\pi_{e}}}\left[l\left(s,\pi \left(s\right)\right)\right]$ However, traditional imitation learning approaches have certain limitations, such as the instability of behavioral cloning and the difficulty of operationalizing reverse reinforcement learning. To address these drawbacks, studies have shown that generative adversarial imitation learning (GAIL) can achieve comparable theoretical and empirical results while being more efficient and avoiding the pitfalls of traditional imitation learning. GAIL employs a GAN framework, where a policy generator G is guided by a reward function represented by a discriminator D. The objective function of GAIL is defined as follows:(6)$$\underset{\pi }{\mathrm{argmin}}\arg\max_{D\in 0,1}E_{\pi }[\log D(s,a)]+E_{\pi_{E}}[\log(1-D(s,a))]-\lambda H(\pi )$$

Here, $H\left(\pi \right)≜E_{\pi }\left[-log\pi \left(a|s\right)\right]$ denotes the entropy of the policy π, and $p_{E}$ represents the joint distribution of experts over state-action pairs. This formulation allows GAIL to derive a policy from expert examples efficiently.

Indeed, GAIL works to improve similarity from generated trajectories to expert trajectories. In this way, the learned strategy is executed in the environment and the update is performed using the gradient descent method. The loss function used for policy updating is as follows:(7)$$\begin{array}{l} l(s,\pi (s))=E[\log D(s,a)]-\lambda H(\pi )≅\\ E_{\tau_{i}}[\log\pi_{a|s}Q(s,a)]-\lambda H(\pi ) \end{array}$$
where $Q(s,a){=}_{\tau_{i}}[log(D(s,a))|s_{0}=s,a_{0}=a]$ represents the state-action value function.

It quantifies the expected log-probability assigned by the discriminator to a given state action pair based on the trajectories $\tau_{i}$. In recent developments, GAIL has demonstrated its effectiveness in environmental reconstruction. This extension leverages the collaborative efforts of multiple agents, allowing for enhanced performance and improved reconstruction outcomes.

In this study, we deploy the SCC-MIE framework to construct a virtual environment incorporating hidden confounders. This is achieved by historical data comprising observable information as well as unobservable confounding factors. By incorporating both types of data, we create a comprehensive and realistic virtual environment to capture the complexity and interactions of the underlying system. We then consider an interactive system with three agents on the basis of GAIL, representing worker policy $\pi_{B}$, platform policy $\pi_{A}$, and confounder policy $\pi_{h}$. We find that worker and platform policies are “mutual environments” from the MDP perspective. The platform’s observations are the workers’ reactions and the platform’s actions are recommendations to the workers. Correspondingly, the observation data of workers is the platform’s recommendation, and their actions are the workers’ responses to the platform. In the environment, hidden confounders generate dynamic effects that affect actions made by the platform and the worker. In other words, these agents interact with each other while all are affected by hidden confounding factors. Specifically, the platform utilizes the workers’ spatiotemporal data to gain the recommendation of workers’ next action via strategy $\pi_{A}$. Then, the worker derives a response for the next temporal state based on the observed data, recommendation, and unobserved confounders via strategy $\pi_{B}$.

Furthermore, we incorporate the dynamic impact of the hidden confounder H by modeling it as a hidden policy denoted as $\pi_{h}$. The main objective of this paper is to leverage the observed data, specifically the trajectories $\left\{x,a_{A},a_{B}\right\}$, to imitate the strategies employed by agents A and B. The influence of the confounders is captured by inferring the hidden strategies. The objective function for multi-agent imitation learning is formulated as follows:(8)$$\arg\underset{\left(\pi_{a},\pi_{b},\pi_{b}\right)}{\min}E_{s∼P_{\tau_{real}}}[L(s,a_{A},a_{B})]$$
where $a_{A}$ and $a_{B}$ are dependent on three policies. According to Equation (7), we apply the GAIL framework to obtain the following:(9)$$\begin{array}{l} L(s,a_{A},a_{B})=E_{\pi_{a},\pi_{h},\pi_{b}}[\log D_{a}(s,a_{A})D_{hb}(s,a_{A},a_{B})-\\ \lambda \sum_{\pi \in \{\pi_{a},\pi_{h},\pi_{b}\}}H(\pi )\\ =E_{\pi_{a}}[\log D_{a}(s,a_{A})]-\lambda H(\pi_{a})+\\ E_{\pi_{h},\pi_{b}}[\log D_{hb}(s,a_{A},a_{B})]-\lambda \sum_{\pi \in \{\pi_{h},\pi_{b}\}}H(\pi )\\ =l(s,\pi_{a}(s))+l((s,a_{A}),\pi_{b}\circ \pi_{h}((s,a_{A}))) \end{array}$$
which demonstrates that the optimization process can be discretized into two components: an optimization strategy $\pi_{a}$ and a joint strategy $\pi_{hb}=\pi_{b}\circ \pi_{h}$.

During the collaborative process, when the reconstruction error stops providing valuable information for the judgment of discriminator D, we then conclude that the reconstructor achieves a satisfactory data reconstruction performance. Thus, Equation (9) essentially minimizes $l\left(s,\pi_{a}\left(s\right)\right)$ and $l(\left(s,a_{A}\right)$, $\pi_{b}\circ \pi_{h}\left(\left(s,a_{A}\right)\right))$, respectively, to output different confidence levels for the data to optimize the reconstruction performance continuously. However, it can be observed that the process of minimizing the loss function increases the reconstruction error for observable samples and decreases the reconstruction error for unobserved confounders. We, therefore, reweight the loss function, i.e., according to the observable data (O) and the unobservable confounding variables (U), using the confidence level.
(10)$$Re-weighting\;factor=\{\begin{array}{ll} & 1+1/p_{i},\;x_{i\in O}\\ & -1/(1-p_{i}),\;x_{j\in U} \end{array}$$

In Equation (10), $p_{i}$ represents the discriminator’s judgment on whether the observation $x_{i}$ is considered observed or not. This equation highlights the distinct behavior of reconstruction errors for observed data and unobserved confounders. Notably, if the discriminator D determines that an observation $x_{i}$ from the set of observed data O is unobservable or unreliable (indicated by a confidence value $p_{i}\to 0$), the corresponding reconstruction error will be noticeably larger. This emphasizes the importance of reliable observations in achieving accurate reconstruction results.

Based on the reweighting function, we materialize the loss function, which is expressed as follows:(11)$$Loss(x)=\left\{ \begin{array}{l} l(s,\pi_{a}(s)),x_{i\in o}\\ l((s,a_{A}),\pi_{hb}((s,a_{A}))),x_{j\in U} \end{array}\right.$$
where
(12)$$l\left(s,\pi_{a}(s)\right)\;≅E_{\tau_{i}}\left[\log\pi_{a}\left(a_{A}|s\right)Q\left(s,a_{A}\right)\right]-\lambda H\left(\pi_{a}\right)$$
and
(13)$$\begin{array}{l} l((s,a_{A}),\pi_{hb}((s,a_{A})))≅E_{\tau_{i}}[\log\pi_{hb}(a_{B}|s,a_{A})Q(s,a_{A},a_{B})]\\ -\lambda \sum_{\pi \in \{\pi_{b},\pi_{b}\}}H(\pi ) \end{array}$$

Moreover, $Q(s,a_{A})=E_{\tau_{i}}[log(D((s,a_{A}))|s_{0}=s,a_{A_{0}}=a_{A}]$ and $Q(s,a_{A},a_{B})=E_{\tau_{i}}[log(D((s,a_{A}),a_{B}))|s_{0}=s,a_{A_{0}}=a_{A},a_{B_{0}}]$ denote the state-action value functions for policy $\pi_{a}$ and policy $\pi_{hb}$, respectively.

By extending the GAIL framework, we accomplish the objective of imitating the strategies employed by each agent. This extension leads to the development of SCC-MIE to address the challenges of hidden confounders in reconstructing the environment using multi-agent methods.

## 5. Detailed Model Construction

In the context of environment reconstruction, we address the problem by introducing a reconstructor and a discriminator, which capture the dependencies within the data to facilitate the reconstruction process. Figure 3 illustrates the workflow.

### 5.1. Confounder Embedded Policy

In our framework, the reconstructor is responsible for inferring the hidden confounders and recovering the unobservable data, while the discriminator evaluates the quality and authenticity of the reconstructed environment. Thus, we assume that the data of agent A and agent B are observable and transparent. However, the data of agent H is unobservable, due to the presence of hidden confounders. As a consequence, we design a joint policy $\pi_{hb}=\pi_{h}\circ \pi_{b}$, which incorporates the confounder policy $\pi_{h}$ and policy $\pi_{b}$. In other words, the joint strategy can be expressed as $\pi_{hb}\left(o_{A},a_{A}\right)=\pi_{b}\left(o_{A},a_{A},\pi_{h}\left(o_{A},a_{A}\right)\right)$, where inputs $o_{A}$, $a_{A}$, and output $a_{B}$ can be obtained from historical data. To summarize, we formulate the environmental reconstruction problem as a Markov decision process, using an imitation learning approach to train both policies by imitating observed interactions.

For the policies update in the generator, the joint policy $\pi_{hb}$ utilizes reward $r_{HB}$, which receives from discriminator D, to update. The policy $\pi_{A}$ updates by reward $r_{A}$. In SCC-MIE, generating the hidden strategy $\pi_{h}$ is a byproduct arising as the policy $\pi_{hb}$ and the policy $\pi_{A}$ are continuously optimized. Such a hidden policy better represents the impact of confounders present in real environments on workers and the platform. Thus, through these two phases, the concealed policy $\pi_{h}$ is iteratively and indirectly optimized to restore the actual confusing impact as much as possible. In order to make the training process faster and more stable, we adopt PPO, which is more efficient in terms of samples than other policy optimization algorithms, to update the above policy.

PPO is a widely used reinforcement learning algorithm that belongs to the policy optimization-based approach. Unlike other algorithms based on policy optimization, its core idea is to optimize the policy while limiting the difference between the new policy and the old one to ensure training stability. To encourage strategy improvement while penalizing excessive strategy updates, a specific objective function is introduced. PPO is adept at handling continuous action spaces and performs well in many tasks without requiring complex tuning parameters. These advantages make PPO an efficient, stable, and general algorithmic alternative for reinforcement learning.

In this framework, we develop a discriminator compatible with both classification tasks for $\pi_{hb}$ and $\pi_{a}$. The discriminator takes as inputs the state-action pairs $\left(o_{A},a_{A},a_{B}\right)$ and $\left(o_{A},a_{A},0\right)$. Firstly, the discriminator classifies the true and generated state-action pairs from $\pi_{hb}$. It then proceeds to classify the state-action pairs from $\pi_{a}$. In contrast to typical generative adversarial learning frameworks where only one task is modeled and learned in the generator, this paper defines the loss functions for $\pi_{hb}$ and $\pi_{a}$ as follows:(14)$$Loss=\{\begin{array}{ll} & loss(\pi_{hb})=E_{\tau }[log(D_{\sigma }(o_{A},a_{A},a_{B}))]+E_{\tau_{real}}[log(1-D_{\sigma }(o_{A},a_{A},a_{B}))]\\ & loss(\pi_{a})=E_{\tau }[log(D_{\sigma }(o_{A},a_{A},0))]+E_{\tau_{real}}[log(1-D_{\sigma }(o_{A},a_{A},0))] \end{array}$$

The probability that the input pairs come from real data is the output to the discriminator. This paper labels the true state-action pairs as 1 and the generated false state-action pairs as 0. The discriminator is trained through supervised learning using these labels. The discriminator is trained for supervised learning by this conception. It is then used as the policy reward while simulating the interaction. The reward function of the policy $\pi_{hb}$ and the policy $\pi_{a}$ can be expressed as follows:(15)$$Reward=\left\{ \begin{array}{l} r^{HB}=-\log(1-D(o_{A},a_{A},a_{B}))\\ r^{A}=-\log(1-D(o_{A},a_{A},0)) \end{array}\right.$$

The confounder-embedded policy plays a crucial role in capturing the influence of hidden confounders, enabling the framework to model and reconstruct the environment accurately. By integrating this policy into the learning process, SCC-MIE aims to achieve robust and accurate reconstruction results, even in unobservable factors.

During the process of environment reconstruction, we begin by formulating the environmental reconfiguration problem as a Markov decision process (MDP) quintuple denoted by (S,A,P,R,γ). In this quintuple, the symbol S represents the state space and represents the action space. The function P:S×A↦S signifies the state transition probability model, and R:S×A↦R denotes the reward function. Additionally, γ serves as the discount factor for cumulative reward. Subsequently, a trajectory is randomly sampled from the observation data, with the initial observation $o_{A}$ assigned as the observation state for the first time slot. To generate a complete trajectory, the policies $\pi_{a}$ and $\pi_{b}$ are employed, triggered by the initial observation $o_{0}^{A}$. Utilizing the policy $\pi_{a}$, the action $a_{t}^{A}$ is determined based on the observation $o_{t}^{A}$. Similarly, the joint strategy $\pi_{hb}$ guides the selection of the action $a_{t}^{B}$. Equation (15) represents simulated rewards used for updating the strategy during the adversarial training phase. Moving forward, given the observation $o_{t}^{A}$ and the action $a_{t}^{B}$, the subsequent observation $o_{t}^{A+1}$ can be obtained using predefined transfer probabilities. This step is repeated until reaching the end state, resulting in the generation of the pseudo-trajectory.

Algorithm 1 elucidates the process of environment reconstruction, utilizing the generative adversarial training framework. The generator takes center stage within each iteration of this algorithm, orchestrating simulated interactions by employing the policy $\pi_{a}$ and the policy $\pi_{hb}$. These interactions lead to the assembly of a trajectory set denoted as $\pi_{sim}$, encompassing the line 5 to 17 of the algorithm. Subsequently, the policy $\pi_{a}$ and the policy $\pi_{hb}$ undergo iterative updates using the proximal policy optimization (PPO) technique, utilizing the generated trajectories $\pi_{sim}$, as a starting point. This transformative process unfolds within line 18 of the algorithm. Guided by the passage of K generator steps, the compatible discriminator assumes its rightful place. In line with the orchestration of Algorithm 1, the compatible discriminator undergoes a two-step training regimen, unfolding within line 20. Notably, the predefined transition dynamics, nestled within line 13, are intricately tied to the specific tasks at hand. These dynamics mold each step of the reconstruction process. SCC-MIE adeptly emulates the observed interaction policies, transcending the confines of mere observation to recover the concealed confounder that lies beyond.
**Algorithm 1:** SCC-MIE algorithm1: **Input:** Trajectory data $D_{real}=\left\{\tau_{1},\tau_{2},\cdot \cdot \cdot ,\tau_{n}\right\}$.2: **Output:** $\pi_{a}$, $\pi_{b}$, $\pi_{h}$.3: Initialization policies $\pi_{hb}$ and $\pi_{a}$ with parameters $\theta_{hb}$ and $\theta_{a}$, and discriminator D with parameter σ;4: **for** i=1, 2, … **do**5:     **for** k=1, 2, …, K **do**6:         $\tau_{sim}=\emptyset$;7:         **for** j=1, 2, …, N **do**8:             $\tau_{j}=\emptyset$;9:             Select a random trajectory $\tau_{r}$ from $D_{real}$ 10:           Set the first state to the initial observation $o_{0}^{A}$;11:           **for** t=0, 2, …, T−1 **do**12:               Simulate the actions $a_{t}^{A}$, $a_{t}^{B}$ by the policy $\pi_{a}$ and the policy $\pi_{hb}$, respectively13:               Calculate the rewards $r_{t}^{A}$ and $r_{t}^{HB}$ by Equation (15);14:               Derive the next observation $o_{t+1}^{A}$ 15:               Insert $\left\{o_{t}^{A},a_{t}^{A},a_{t}^{B},r_{t}^{A},r_{t}^{HB}\right\}$ into the trajectory $\tau_{j}$ 16:           **end for**17:           Integration of the computed data18:           **end for**19:           Update parameters $\theta_{hb}$ and $\theta_{a}$ according to PPO;20:   **end for**21:   Update the discriminator D by minimizing the losses;22: **end for**

### 5.2. Worker Selection

In the dynamic real world, worker participation is real time and random. Faced with unknown information, this paper decides whether to select it or not based on the importance (confidence) of the data, i.e., workers who contribute high-quality data are selected and workers with low-quality data are discarded. In addition, we simplify the worker selection problem by viewing it as choosing the best one out of w workers. In addition, we simplify the worker selection problem by viewing it as choosing the best one out of w workers. In this case, the worker selection problem is a typical secretarial problem. To better solve this problem, we first construct a sample set of workers and observe and eliminate the top 1/e workers to understand their utility distribution. Then, in practice, we use historical data to learn about this distribution in real time and then select the best worker from the known distribution.

To streamline the process and harness the potential of forthcoming workers, we introduce a novel sampling methodology, aptly named Algorithm 2. This ingenious approach enables us to make informed decisions by capitalizing on historical data. The algorithm begins by extracting a sample set of size j−1 from the wealth of past records (line 1). Subsequently, we meticulously evaluate the utility of each upcoming worker against the pinnacle of utility, also known as the threshold, within the sample set. This critical assessment dictates whether a worker is selected for further consideration (line 3). When the current worker is not chosen, they are gracefully assimilated into the sample set, seamlessly replacing one of its members through a random resampling process (line 5–line 6). Importantly, it should be noted that the threshold evolves dynamically over time as we progressively eliminate workers with the highest utility. Consequently, our endeavor entails creating a sample set comprising j−1 workers, emulating a near approximation of the discarded workforce. This astute strategy allows us to gain valuable insights into estimating the threshold for every new worker, laying the foundation for informed decision making.
**Algorithm 2:** Worker selection algorithm1: **Input**: selected workers μ, new workers $\omega_{i}$, time: T; budget: B2: Conduct X from history data according to t and T.3: **while** $c_{i}<=B$ **do**4:       if $D(\mu ,\omega_{i})>=\max\{X\}$ **then**          return $\mu \cup w_{i}$;5:       **else**6:           $X\cup D(\mu ,\omega_{i})$;7:           Select z at random from X, X=X | {z};8:           $\omega_{i}++$ 9:       **end if**10: **end while**

## 6. Performance Evaluation

The AIMSUN dataset is a virtual collection of vehicle trajectories that has been constructed using a microscopic traffic simulation model. The system utilizes a dynamic traffic assignment method to suggest appropriate routes for individual vehicles [25]. Next, we analyze five distinct models: binomial, C-logit, proportional, polynomial logit, and fixed. The several route models in the AIMSUN dataset offer alternative methods for determining probability and making judgements when selecting routes for vehicles. The fixed model employs a greedy approach that prioritizes minimizing the trip time for each origin–destination (OD) pair. By examining these route models, one can assess the performance of SCC-MIE in comparison to various tactics. This evaluation offers valuable insights into the usefulness of SCC-MIE and allows for a comparison with the baseline models in addressing route selection difficulties in the AIMSUN dataset.

Table 2 presents a partial set of data, and Figure 4 is a network diagram of three routes based on the data in Table 2.

This section provides the training and testing outcomes of SCC-MIE and baseline models for AIMSUN. The performance review encompasses several facets and diverse performance criteria. In order to guarantee the ability to replicate the results, the specific parameters utilized throughout the training procedure are outlined in Table 3.

### 6.1. Baseline

**Mobility Markov Chain [26]**: Mobile Markov chain (MMC) is a fundamental model commonly used to address location prediction problems. MMC models represent a stochastic process where transitions occur between different states in the state space. One key characteristic of MMC models is their “memoryless” nature, meaning that the probability distribution of transitioning to the next state is solely determined by the current state.**Recurrent Neural Network [27]**: Recurrent neural network (RNN) is a popular and robust algorithm for location recommendation tasks. RNN models excel at capturing the spatiotemporal characteristics in data, allowing them to make accurate predictions for the next location. In this study, we employ long short-term memory (LSTM) for modeling continuous data.**Inverse Reinforcement Learning [28]**: Inverse reinforcement learning (IRL) aims to infer unknown reward functions based on observed demonstrations or expert behaviors in order to train RL agents or guide their decision-making process in new, unknown environments. In this study, we employ maximum entropy IRL (MaxEnt) to extend the idea of matching state visits to matching state-action visits.**Generative Adversarial Imitation Learning [6]**: GAIL is an imitation learning algorithm based on generative adversarial networks (GAN), which allows a learner to learn strategies to imitate experts by confronting them.**Deconfounded Multi-agent Environment Reconstruction [17]**: DEMER uses a multi-agent generative adversarial imitation learning framework. It is proposed to introduce a confounder embedding strategy and train the strategy using a compatible discriminator.

### 6.2. Result

This section presents the training and testing results of the model. It begins with the convergence curves for the training results, followed by the accuracy of trajectory similarity. Next, it compares the accuracy of expert and learner models based on imitation learning. Finally, it compares the running time of each model.

**Convergence Curve:** Figure 5 illustrates the convergence curve of the loss functions (Discrim_loss,Policy_loss,Value_loss) and the causal entropy $\left(H(\pi_{\theta })\right)$ based on the AIMSUN dataset. From Figure 5, we find that when the number of iterations is small, both discrim_loss and Value_loss decrease, while policy_loss increases. For example, when the number of iterations starts, Discrim_loss is 1.38, Policy_loss is −1.44, and Value_loss is 0.49. When the number of iterations is increased to 20, the changes in each loss function are more obvious, i.e., Discrim_loss shows a steep drop to 0.53, while Policy_loss decreases to −0.93, and Value_loss decreases to 0.14. As the number of iterations is 30, Discrim_loss is 0.12, Policy_loss is −0.14, Value_loss is 0.0015. As the number of iterations is 40, Discrim_loss is 0.08, Policy_loss is −0.11, Value_loss is 0.00011. This is because the policy generator and the discriminator have different design purposes in the framework. The main goal of the policy generator is to learn to generate action sequences that are similar to the expert’s policy by maximizing Policy_loss in order to achieve high-quality trajectory generation. In contrast, the discriminator is designed to minimize the gap between the action sequences generated by the generator and the real samples so that it can accurately distinguish between the two. In addition, the framework introduces the concept of confidence level so that the SCC-MIE discriminator can guide the training process of the generator more effectively. When the generator returns a high confidence level, the discriminator has a lower probability of misclassifying the generated action sequences. Thus, the loss function of the discriminator decreases rapidly, which motivates the generator to generate trajectories closer to the expert’s strategy. This mechanism accelerates the training process and improves the efficiency of the algorithm.

However, as training continues, the discriminator begins to discriminate real trajectories from the generated trajectories to the point where the accuracy of the generator continues to decrease. Therefore, in the middle of iterations, discrim_loss tends to increase, and policy_loss tends to decrease. For example, when the number of iterations is 200, we can observe that discrim_loss = 1.07, policy_loss = −0.59. As the number of iterations increases to 500, we find that discrim_loss is 1.32 and policy_loss is −0.63. When the number of iterations varies to 10,00, we find that discrim_loss is 1.38 and policy_loss is −0.66.

In addition, we find that value_loss is decreasing and almost converges to zero for the increasing number of iterations. For example, when the number of iterations is 1000, value_loss = 0.00054. At this point, the entropy $H_{\pi_{\theta }}$ converges to the maximum value of causal entropy. To ensure that the results are correct, we increased the number of training times to 10,000 and realized that none of the values had changed significantly. This is because as the number of iterations increases, all the values reach convergence to the point at which the discriminator is unable to distinguish the real trajectory from the generated trajectory. Therefore, SCC-MIE achieves the best trajectory generation through environment reconstruction.

**Accuracy of Trajectory Similarity:** Figure 6 depicts the precision of the trajectories produced by each model over the five route models. We employed two widely utilized assessment metrics in sequence modelling to evaluate the similarity at the trajectory level: the BLEU [29] score and the METEOR [30] score. Both ratings have a maximum value of 1, while higher values indicate superior accuracy.

The findings indicate that DEMER and SCC-MIE exhibit considerably higher levels of accuracy across all five route models. As an illustration, DEMER achieves a mean BLEU score of 0.9931 and a mean METEOR score of 0.9887, whereas SCC-MIE attains a mean BLEU score of 0.9986 and a mean METEOR score of 0.9985. Conversely, the IRL method has the poorest performance overall, as seen by its lowest BLEU score of 0.6843 in the C-logit model and its lowest METEOR score of 0.4724 in the binomial model.

**Accuracy of Expert and Learner:** Figure 7 exhibits that the accuracy of the expert and the learner is different for the number of training iterations. In this experiment, we divide the entire dataset into a training dataset and a test dataset in the ratio of 0.7:0.3. The number of training iterations is dynamically varied in the range [10, 10,000]. Then, all models use 1000 sample trajectories to compare the model’s performance properly. Table 3 provides further information regarding the hyperparameters of SCC-MIE. The learner’s accuracy is shown to improve with an increasing number of training rounds, as depicted in Figure 7. For example, the number of iterations is 10, the accuracy of the expert is 100%, and the accuracy of the learner is 51.54%. When the number of iterations is increased to 100, the accuracy of the expert is 85.68%, whereas the accuracy of the learner is 84.43%. The accuracy of novice learners is expected to be poorer due to their incomplete understanding of the expert’s method.

In addition, the learner’s accuracy shows large fluctuations because he/she is still trying to understand and imitate the expert’s strategy. As the training progresses, the learner gradually improves his/her strategy, and the accuracy increases. More often than not, the learner will gradually converge to a strategy that approaches or exceeds the expert’s, thus stabilizing accuracy and eventually meeting or exceeding it. For instance, the number of iterations is 10,000, the accuracy of the expert is 96.76%, and the learner’s accuracy is 96.29%. The number of iterations is 100,000, the accuracy of the expert is 98.42%, and the learner’s accuracy is 99.42%.

Therefore, the model training relies on expert experience at first. Then, as the number of training sessions increases, the learner gains experience, and the accuracy rate continues to rise.

Moreover, we can see that the accuracy of learners in GAIL is highly variable from Figure 7. For example, when the number of iterations is 10, the expert achieves a perfect accuracy of 100%, whereas the learner’s accuracy is 54.92%. When the number of iterations increases to 100, the expert’s accuracy is 85.75%, while the learner’s accuracy is 86.26%. Due to the limitations of real environments in providing complete and observable information, the reconstruction of the environment is hindered.

DEMER attempts to reconstruct hidden confounding factors to better structure the environment. The curve of the learner in DEMER follows a similar trend to the curve in the other models. For example, when the number of iterations is 10, the expert reaches a perfect accuracy of 100% and the learner’s accuracy is 69.7%. When the number of iterations is increased to 100, the expert’s accuracy is 85.71% and the learner’s accuracy is 71.88%. The DEMER model provided inspiration in providing complete and observable information, so we developed a new approach called SCC-MIE. This approach improves the performance of GAIL (generative adversarial imitation learning) and DEMER (deconfounded multi-agent environment reconstruction) and demonstrates the beneficial effects of our confounding factor setting.

**Running time:** Figure 8 depicts the computation time, measured on a GPU, required to generate 10,000 vehicle trajectories using six different models. Interestingly, it is observed that all six models exhibit the capability to generate the desired 10,000 vehicle trajectories in less than 2 s. Figure 9 (SCC-MIE) demonstrates that the running times of the six approaches are relatively similar when the number of iterations reaches 1000.

In general, when the number of iterations is comparatively lower, the above trajectory generation algorithm may have a relatively shorter running time. However, virtual environments in large trajectory recommendation tasks are highly dynamic and hybrid, requiring a large number of iterations to achieve good results. When the number of iterations keeps increasing, the running time of algorithms, such as MMC, IRL, GAIL, DEMER, etc., may show a nonlinear increase because of the complexity of the environment construction and the high dimensionality of the state space. In contrast, the increase in SCC-MIE is relatively small and the application value is high.

## 7. Conclusions

In this paper, we proposed self-interested coalitional crowdsensing for multi-agent interactive environment monitoring (SCC-MIE), based on the generative adversarial training framework to construct a virtual sensing environment with hidden confounders to conduct the environment monitoring in real-time. To begin with, we incorporated the confounder-embedded policy into the generator and ensured compatibility of the discriminator with various classification tasks, thus facilitating precise optimization of each strategy. After that, we introduced a novel self-interested coalitional learning scheme that can cooperate with additional discriminators to provide interpretable confidence. The confidence level obtained from this module can be utilized for worker selection, ensuring that the chosen workers contribute more meaningful data for reconstructing the sensing environment. Finally, we evaluated the framework using an AIMSUN-based trajectory dataset. Extensive experiments against state-of-the-art baselines have demonstrated the effectiveness and robustness of our approach.

## Figures and Tables

**Figure 1 sensors-24-00509-f001:**
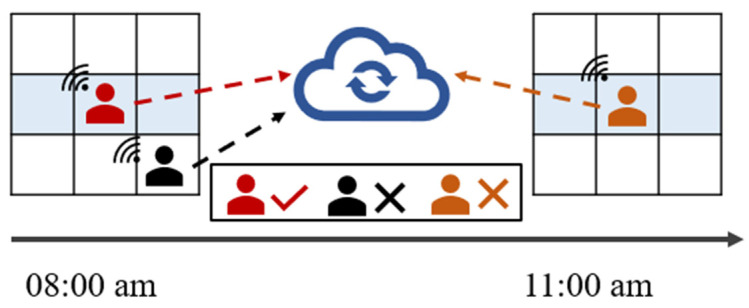
Data collected at different times in different regions.

**Figure 2 sensors-24-00509-f002:**
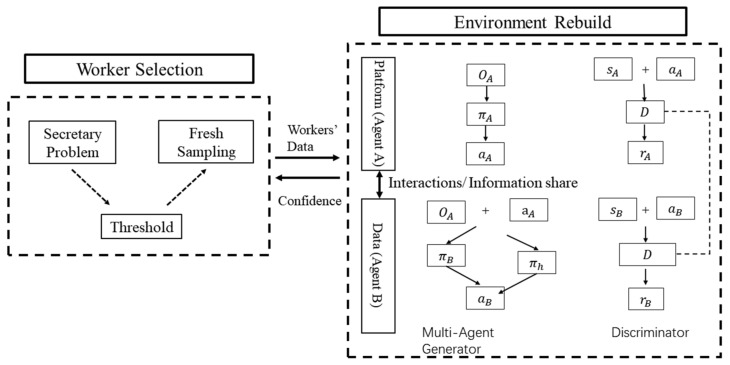
System framework SCC-MIE.

**Figure 3 sensors-24-00509-f003:**
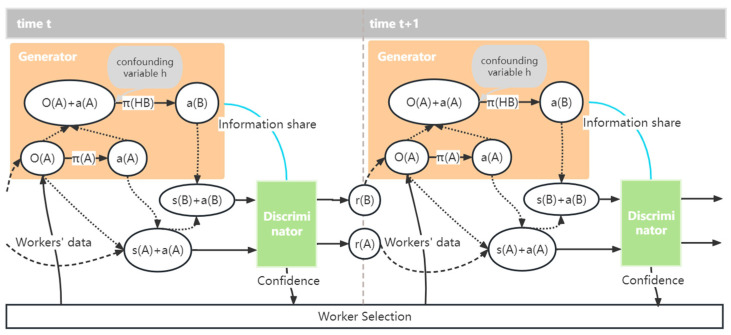
Workflow diagram of SCC-MIE.

**Figure 4 sensors-24-00509-f004:**
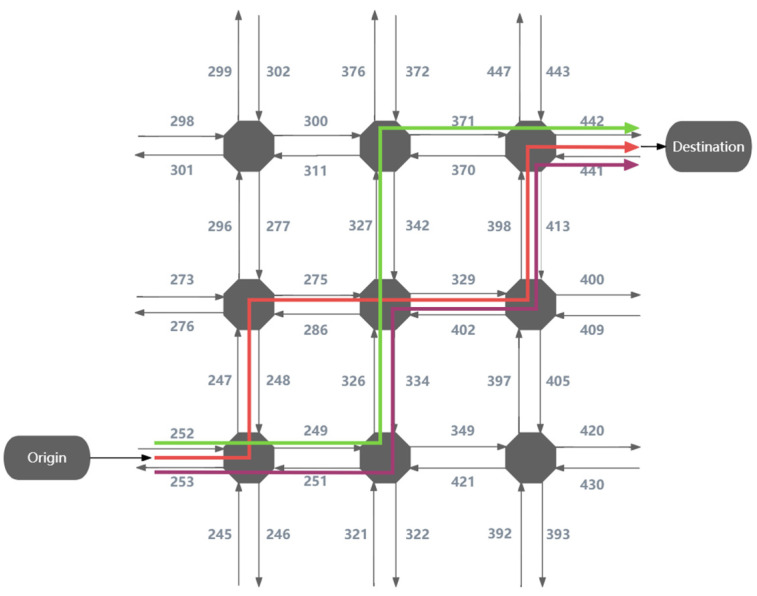
Simulated transport network in an AIMSUN environment. The red line represents the trajectory associated with OriginID = 1, the purple color represents the trajectory associated with OriginID = 2, and the green color represents the trajectory associated with OriginID = 3.

**Figure 5 sensors-24-00509-f005:**
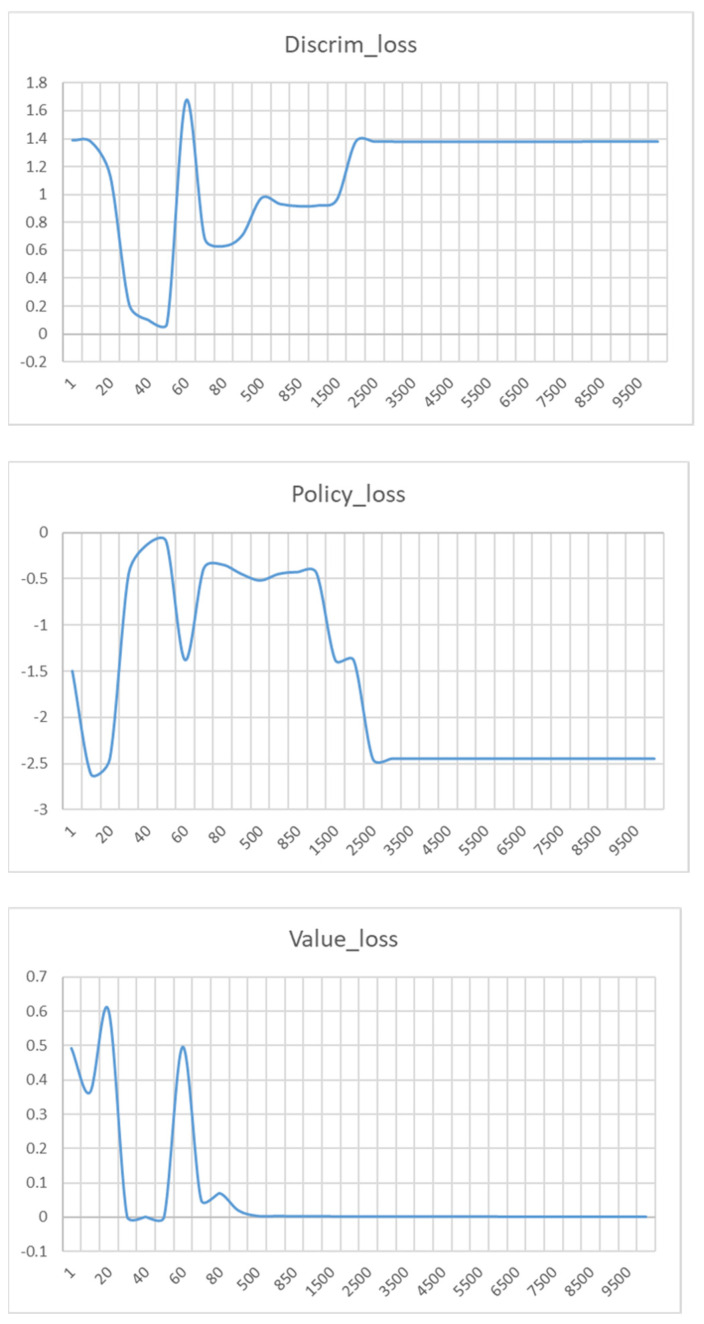
Loss functions of Discrim, Policy, Value, and Entropy.

**Figure 6 sensors-24-00509-f006:**
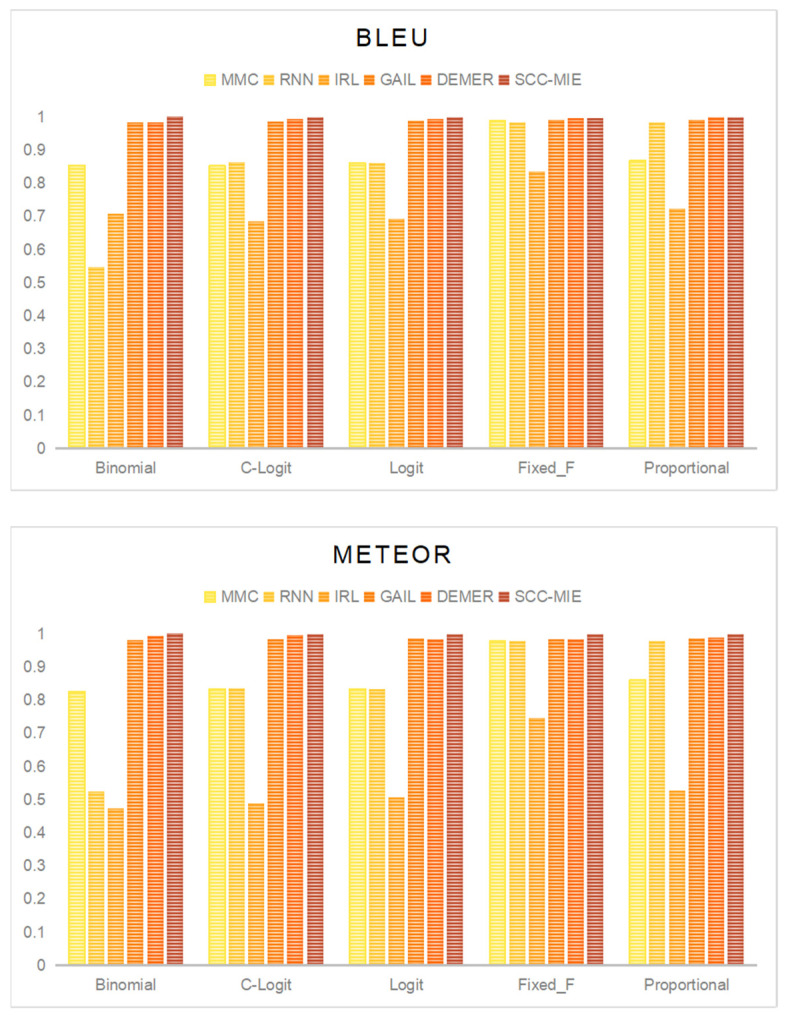
Accuracy of trajectory similarity.

**Figure 7 sensors-24-00509-f007:**
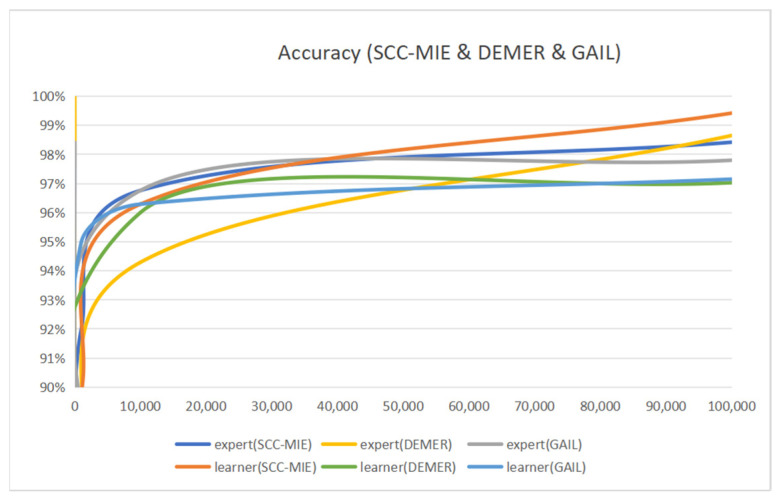
Accuracy of SCC-MIE, DEMER, and GAIL.

**Figure 8 sensors-24-00509-f008:**
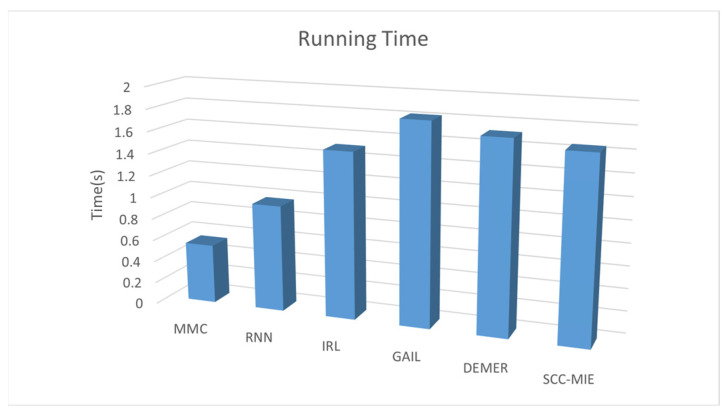
Running time taken to generate 10,000 vehicle trajectories.

**Figure 9 sensors-24-00509-f009:**
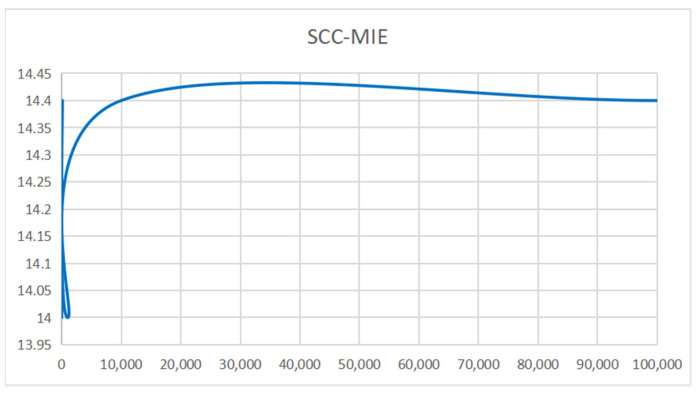
Running time of SCC-MIE.

**Table 1 sensors-24-00509-t001:** List of important notations.

Notation	Explanation
i,N	Index of a worker, total of workers
t,T	Index of time slot, total of time slot
$X_{i}^{t}$	Sensory data of worker $w_{i}$ at t-time slot
$\hat{X}$	Reconstructed data for X
O,U	Observable data set and unobservable data set
$\tau =\{\tau_{1},\tau_{2},\ldots ,\tau_{n}\}$	Sensing the trajectory of workers.
$c_{i}$	Cost of collecting data.
B	Budget of collecting data.
$P,p_{i}$	Confidence level, the judgment of the discriminator D

**Table 2 sensors-24-00509-t002:** Simulated traffic network partial data in AIMSUN environment.

OriginID	Number	SectionID
1	1	252
1	2	247
1	3	275
1	4	329
1	5	398
1	6	442
2	1	252
2	2	249
2	3	326
2	4	329
2	5	398
2	6	442
3	1	252
3	2	249
3	3	326
3	4	327
3	5	371
3	6	442

**Table 3 sensors-24-00509-t003:** Hyperparameters used for test.

Hyperparameter	Value
Number of iterations	10,000
Number of episodes	10,000
${\mathrm{Batch}}_{\mathrm{size}}$	2048
Number of hidden neurons	64
Learning rate	0.00003
Discount rate of reward	0.99
Entropy coefficient	0.01

## Data Availability

Data are contained within the article.

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
