# Peer review of "Self-Interested Coalitional Crowdsensing for Multi-Agent Interactive Environment Monitoring"

_sensors, 2024, doi:10.3390/s24020509_

Round 1

Reviewer 1 Report

Comments and Suggestions for Authors

This paper introduces a new framework called Self-Interested Coalitional Crowdsensing for Multi-Agent Interactive Environment Monitoring (SCC-MIE), and presents methods to enhance the efficiency and accuracy of environmental monitoring by addressing heterogeneous and sparse data, as well as hidden causal factors.

This paper clearly explains the various equations and algorithms applied in the proposed framework, and they mostly seem valid. However, despite the abundance of proposed formulas, algorithms, and related studies, their performance evaluation appears relatively insufficient. To more clearly highlight the advantages of the framework, there should be performance measurements and comparative evaluations not only of the proposed algorithms but also of related research and existing algorithms. Therefore, it is requested to add comparisons with other frameworks or systems based on the performance measurement figures of the current proposed framework.

Reviewer 2 Report

Comments and Suggestions for Authors

Further validations in diverse real-world scenarios could strengthen the method.

The authors need to asses the scalability of the proposed model in various environments

A more detailed comparison with state-of-art could provide a clear understanding of the advantages the method brings.

Comments on the Quality of English Language

Pozitive aspects:

The language used is clear and precise, effectively communicating complex technical concepts

Transitions between sections and ideas appear to be smooth, contributing to an easy-to-follow narrative

Improvements:

Simplifying some of the more complex sentences could enhance understanding for a broader audience.

Overuse of passive constructions can make the text seem impersonal and can sometimes obscure the clarity of statements.

Some sections may have lengthy paragraphs that pack multiple ideas or concepts. These paragraphs could be broken into shorter parts.

Reviewer 3 Report

Comments and Suggestions for Authors

1. Fig 5, what is the unit for time?

2. What are the environment variables considered for reinforcement learning? 

3. How values or table for reinforcement learning is maintained? whether Q table is maintained? 

4. What is meant by discount rate of reward?

5. Fig 4, combined GAIL & SCC-MIE chart also can be added for better visibiliy of two model comparison.

6. How multi-agents are maintained? Why multi-agent is needed?

7. Some details or sample data from AIMSUM dataset can be added

8. Proposed work flow diagram can be added.

Round 2

Reviewer 3 Report

Comments and Suggestions for Authors

author made changes based on review comments